# Supplementation with Antimicrobial Peptides or a Tannic Acid Can Effectively Replace the Pharmacological Effects of Zinc Oxide in the Early Stages of Weaning Piglets

**DOI:** 10.3390/ani13111797

**Published:** 2023-05-29

**Authors:** Limin Tan, Yuyue Xi, Chengyu Zhou, Yetong Xu, Jiaman Pang, Xie Peng, Zhiru Tang, Weizhong Sun, Zhihong Sun

**Affiliations:** Laboratory for Bio-Feed and Molecular Nutrition, College of Animal Science and Technology, Southwest University, Chongqing 400715, China; tanlimin1997@foxmail.com (L.T.); xiyuyue0202@163.com (Y.X.); zhouchengyu1390@163.com (C.Z.); xyt8501@163.com (Y.X.); pangjm@swu.edu.cn (J.P.); pengxie2022@swu.edu.cn (X.P.);

**Keywords:** zinc oxide, tannic acid, antibacterial peptide, piglets, intestinal microorganisms

## Abstract

**Simple Summary:**

In the past, zinc oxide was often used to control diarrhea in weaned piglets; however, it also resulted in a serious environmental burden. It is critical to identify safe and efficient substitutes for zinc oxide to prevent and treat diarrhea in weaned piglets. This study showed that tannic acid or antimicrobial peptides had better effects added to the feed as substitutes for zinc oxide to ensure the intestinal health of weaned piglets.

**Abstract:**

Zinc oxide (ZnO) harms the environment and can potentially increase the number of drug-resistant bacteria. Therefore, there is an urgent need to find safe and effective alternatives to improve gut health and reduce the incidence of diarrhea in weaned piglets. This study conducted an antibacterial test of ZnO, antibacterial peptides (AMPs), and tannic acid (TA) in vitro. Thirty piglets were randomly allotted to one of the following three dietary treatments: ZnO (2000 mg/kg ZnO diet), AMPs (700 mg/kg AMPs diet), and TA (1000 mg/kg TA diet). The results showed that the minimum inhibitory concentrations of ZnO and TA against *Escherichia coli* and *Salmonella* were lower than those of AMPs, and the minimum inhibitory concentrations of ZnO, AMPs, and TA against *Staphylococcus aureus* were the same. Compared to ZnO, AMPs increased the digestibility of dry, organic matter and the crude fat. Additionally, TA significantly (*p* < 0.05) increased the digestibility of dry and organic matter. On experimental day 14, the plasma interleukin-6 (IL-6) content of piglets supplemented with AMPs and TA was increased significantly (*p* < 0.05). On experimental day 28, alanine aminotransferase activity in the plasma of weaned piglets in the ZnO and TA groups was significantly (*p* < 0.05) higher than in piglets in the AMPs group. The levels of plasma IL-6 and immunoglobulin M (IgM) were significantly higher (*p* < 0.05) in the ZnO and AMPs groups than in the TA group. On experimental days 14 and 28, no significant differences were observed in the antioxidant capacity among the three experimental groups. Intestinal microbial diversity analysis showed that the Chao1 and ACE indices of piglets in the AMPs group were significantly higher (*p* < 0.05) than those in the ZnO and TA groups. At the genus level, the relative abundance of *Treponema_2* was higher in the feces of piglets fed a diet supplemented with TA than in those fed diet supplemented with ZnO (*p* < 0.05). The relative abundance of *Lachnospiraceae* was higher in the feces of piglets fed a diet supplemented with AMPs than in those fed diet supplemented with ZnO or TA. Overall, AMPs and TA could be added to feed as substitutes for ZnO to reduce diarrhea, improve nutrient digestibility and immunity, and increase the abundance of beneficial intestinal bacteria in weaned piglets.

## 1. Introduction

Alterations in pig feed and the external environment result in an extremely unstable state of health in weaned piglets [1]. Piglets do not have a mature intestinal structure, and solid food particles cannot be fully digested. Fermentation of undigested foodstuffs promotes the growth of bacterial pathogens in the intestinal tract, causing an imbalance in gut flora, a major cause of diarrhea in weaned piglets [2]. Diarrhea can cause disorders of the intestinal system and decrease immunity in piglets. This leads to high mortality, low feed intake, and weight gain, which severely affects piglet growth [3,4]. Zinc oxide (ZnO), given at a pharmacologic dose of 1500‒2500 mg/kg of feed, has antibacterial and growth-promoting properties that increase the body’s antioxidant capacity and reduces the occurrence of withdrawal stress. The long-term use of a high amount of ZnO will increase the proportion of drug-resistant bacteria in the piglet [5,6,7]. However, the digestibility of ZnO in animal feed is extremely low, and its excretion and excreta affect the soil ecosystem and the fundamental functions of the soil [8]. Since June 2022, the European Commission has prohibited the use of pharmacological doses of ZnO [9]. Thus, there is an urgent need for novel feed additives or methods to replace pharmacological doses of ZnO.

The most commonly investigated alternatives are feed additives such as organic acids, probiotics, or original plant formulas. Antimicrobial peptides (AMPs) are small molecule (<10 kDa) host defense substances that are commonly found in both invertebrate and vertebrate innate immune systems [10,11]. As an important component of the innate immune system, AMPs provide effective, non-specific defenses against infections [11]. AMPs exert multiple antimicrobial activities that may provide a strategy for preventing bacteria from acquiring resistance [12]. In addition to directly attacking microbes, AMPs may confer protection through alternative mechanisms such as maintaining normal gut homeostasis and modulating host inflammatory responses [13]. AMPs have effectively improved the growth performance, intestinal morphology, and immunity of weanling piglets and reduced harmful gut microbiota [14,15]. Tannic acids (TA) are a group of ubiquitous polyphenols in plant organs [16]. From a chemical perspective, TA can be classified into hydrolyzable tannins, which can be hydrolyzed to gallic acid and glucose, and condensed tannins, which are flavonoids [17]. Certain amounts of TA are considered anti-nutritional factors as they can cause the precipitation of proteins, inhibit digestive enzymes, and negatively affect the utilization of vitamins and minerals [18]. Recent studies have revealed that TAs benefit monogastric animals if used cautiously [19,20,21]. The beneficial effects of TA in swine husbandry are related to its antibacterial, antiviral, antioxidant, and free radical scavenging and anti-inflammatory activities [18,22]. TA forms an insoluble denatured protein film that coats the intestinal mucosal surface. This membrane may prevent bacteria from adhering to the gut epithelium and inhibit the growth of harmful bacteria [23]. Thus, adding a certain amount of TA to the diet of weanling piglets may regulate intestinal flora, reduce the occurrence of post-weaning diarrhea, and improve growth performance [24,25,26].

This experiment used ZnO, AMPs, and TA as the different groups to compare growth performance, nutrient digestibility, gut microbes, and blood biochemistry of weaning piglets. We investigated whether AMPs and TA could effectively replace the effects of pharmacological doses of ZnO on weaned piglets in the same growth environment.

## 2. Materials and Methods

### 2.1. Ethics Approval

All experimental procedures were approved by the Southwest University Animal Care and Use Committee (ethical license number: IACUC-20210310-10).

### 2.2. Antibacterial Test In Vitro

#### 2.2.1. Suspension Preparation

Stock suspensions were prepared by resuspending ZnO in sterile water to a final concentration of 8 mg/mL. The suspension was maintained at 4 °C and set aside after mixing by vigorous vortexing. AMPs and TA suspensions were prepared using the same procedure.

#### 2.2.2. Bacterial Culture Conditions and Antibacterial Test

The standard strains of *Escherichia coli* (CVCC 1570), *Staphylococcus aureus* (CVCC 1882), and *Salmonella* (CVCC 3374) used in the antibacterial test were obtained from Professor Zhiru Tang (College of Animal Science and Technology, Southwest University). The species were previously identified using real-time PCR, high-resolution melting analysis, and sequencing methods.

The bacterial liquid concentration of the activator was adjusted to 106 CFU/mL for standard antimicrobial assays. Three bacterial indicator suspensions (100 μL) were spread evenly on a solid agar plate, and four sterile Oxford cups were gently placed into a Petri dish with forceps. Sterile water treatment was performed with a blank control group with five gradients of bacterial suspension concentration (8000, 4000, 2000, 1000, and 500 mg/mL). Then, 200 μL of the individual concentration gradient suspension was added to the corresponding Oxford cup. After 24 h of culturing, the antimicrobial effect was determined by measuring the size of the antimicrobial ring.

### 2.3. Animals and Experimental Design

Experiments were performed on 4-week-old weaned piglets (Duroc × Yorkshire × Landrace). Piglets were individually housed in stainless steel cages (1.80 m length × 1.20 m height × 0.90 m width) equipped with water nipples offering free access to water. The room temperature was maintained at 24.0–26.0 °C.

Thirty piglets were randomly assigned to one of the following three dietary treatments: ZnO (2000 mg/kg of ZnO diet), AMPs (700 mg/kg of AMPs diet), and TA (1000 mg/kg of TA diet). Three additives were provided by Sichuan zoological academy, the main component of AMPs was derived from animal intestinal active substances, and the main component of TA was 50% hydrolyzed tannins. Each dietary treatment had 10 repetitions with one pig each. The experimental period lasted 35 days, including an adaptation period for the first 7 days of the experiment. The base diet was formulated following the recommendations of the National Research Council [27]. The ingredients and chemical compositions of the diets are presented in Table 1. The daily feed intake (45 g/kg) was based on the recommendation of the National Research Council [27] and was provided as three equal meals at 08:00, 12:00, and 18:00.

### 2.4. Measurements and Sampling

The amounts of feed offered and refused were recorded daily. The piglets were weighed at the start (day 1) and end (day 29) of the experiment. Fecal indices of individual piglets were recorded throughout the experiment. The fecal consistency score of piglets was determined according to the following criteria: 0, fecal formation, strip or granular; 1, soft feces, visible; 2, viscous, unformed, high moisture content; 3, liquid, amorphous, fecal water separation. When fecal consistency was scored as 2 or 3, the piglet was considered to have diarrhea. For each piglet, the diarrhea incidence was calculated as follows: number of days with diarrhea/28 × 100. From days 22 to 28, the feed was supplemented with 0.3% TiO_2_ as a digestibility marker. Feces (approximately 0.1 kg per pig per collection) were collected at 08:00, 15:00, and 23:00 from days 26 to 28. The collected feces were combined with 10% sulfuric acid (10 mL/100 g feces) to prevent evaporation and loss of nitrogen and frozen at −20 °C for later chemical composition analysis. On days 14 and 28, six piglets from each group were randomly selected for blood sample collection. Five mL of a blood sample from the anterior vena cava were collected by jugular puncture into 10-mL tubes treated with sodium heparin. The plasma was harvested by centrifugation at 3000× *g* for 20 min and then stored at −80 °C for later analysis. On day 28, six piglets from each group were randomly selected for fresh feces collection (approximately 20 g per pig per collection). The collected feces were frozen in liquid nitrogen and stored at −80 °C for 16 S rRNA analysis.

### 2.5. Calculation and Chemical Analysis

The feed and feces’ dry matter, CP, calcium, phosphorus, organic matter, crude ash, and crude fat contents were determined using previously described methods [28]. 

Based on the initial weight, final weight, and feed intake, growth performance was calculated according to the following formula: average daily weight gain (ADG) = total weight gain (g)/days (d); average daily feed intake (ADFI) = total feed intake (g)/days (d), and ratio of feed intake to body weight gain (F/G) = feed intake/weight gain.

The concentrations of immunoglobulins (IgA, IgM, and IgG), interleukin (IL-2, IL-4, IL-6), and growth hormone (GH) in the plasma were determined using ELISA kits from Wuhan Merck Biotechnology Co., Ltd., Wuhan, China). Alkaline phosphatase (ALP), aspartate aminotransferase (AST), glutamic pyruvic transaminase (ALT), total antioxidant capacity (T-AOC), total superoxide dismutase (T-SOD), catalase (CAT), malondialdehyde (MDA), and glutathione peroxidase (GSH-PX) levels were determined using commercial kits from Nanjing Jiancheng Co., Ltd. (Nanjing, China). 

The microbial 16 S rRNA sequencing of the fecal samples was performed by Biomarker Technologies Co., Ltd. (Beijing, China). Microbial diversity was based on the Illumina NovaSeq sequencing platform, sequenced using the paired-end method to create a small fragment library. The species composition of samples was revealed by splicing, filtering, clustering, denoising, species annotation, and abundance analysis. Further, alpha and beta diversity analyses, dominant species diversity analysis, correlation analysis, etc., were conducted.

### 2.6. Statistical Analysis

The data were analyzed by one-way analysis of variance (ANOVA) using SPSS statistics 26 software. The Duncan method was used for multiple comparisons of differences between groups, and the results were expressed as the mean and standard error (SEM). When *p* < 0.05, the difference was considered significant.

## 3. Results

### 3.1. Minimum Inhibitory Concentration

Table 2 shows that the minimum inhibitory concentration (MIC) of ZnO against *E. coli*, *S. aureus*, and *Salmonella* was 1 mg/mL. The MIC of AMPs against *E. coli*, *S. aureus*, and *Salmonella* were 4, 1, and 2 mg/mL, respectively. The MIC of TA against *E. coli*, *S. aureus*, and *Salmonella* were 1, 1, and 0.5 mg/mL, respectively. 

### 3.2. Growth Performance

There were no significant (*p* > 0.05) differences in the average daily gain, average daily feed intake, and feed intake/body weight gain ratio of piglets fed diets with ZnO, AMPs, or TA (Table 3). The diarrhea rate of piglets in the three groups was not significantly (*p* > 0.05) different. Compared with that of the ZnO group, the digestibility of dry and organic matter of piglets in the diet supplemented with AMPs and TA was significantly (*p* < 0.05) higher. The crude fat digestibility of piglets in the TA group was significantly (*p* < 0.05) higher than that in the ZnO group, but no significant (*p* > 0.05) difference was found between the TA and AMPs groups.

### 3.3. Blood Parameters

On experimental day 14, no significant (*p* > 0.05) differences were observed in plasma ALP, AST, and ALT activities or plasma GH content among the three groups (Table 4). On experimental day 28, the plasma ALT activity of weaned piglets in the ZnO and TA groups was significantly (*p* < 0.05) higher than that of piglets in the AMPs group, and the plasma GH content of piglets in the ZnO group was significantly higher (*p* < 0.05) than that of pigs in the TA groups.

On experimental day 14, the levels of immune proteins (IgA, IgM, and IgG) in the plasma of piglets in the three groups were not significantly (*p* > 0.05) different (Table 5). On experimental day 28, the plasma IgM content of piglets in the ZnO and AMPs groups was higher (*p* < 0.05) than that of piglets in the tannic acid group. 

Compared to that of the ZnO group, the plasma interleukin-6 content of piglets in the AMPs and TA group increased significantly (*p* < 0.05) on experimental day 14 (Table 6). The plasma content of IL-6 in piglets fed the diet with TA was significantly (*p* < 0.05) lower than that in piglets fed the diet with ZnO and AMPs on experimental day 28. There were no significant (*p* > 0.05) differences in the plasma IL-2 and IL-8 levels among the three groups throughout the test period. 

The results for the plasma antioxidant capacity of piglets are presented in Table 7. There were no significant (*p* > 0.05) differences in plasma T-AOC, T-SOD, CAT, MDA, or GSH-PX activities among the three trial groups.

### 3.4. Fecal Microbiota 16S rRNA Gene Analysis

As shown in Figure 1a, the number of fecal microorganisms shared by the three groups of piglets was 743. The alpha diversity showed that there were no significant (*p* > 0.05) differences in the Shannon and Simpson indices of fecal microorganisms among the three groups, but the Chao1 and ACE indices of fecal microorganisms of the piglets in the AMPs group were significantly (*p* < 0.05) higher than those of the piglets in the ZnO and TA groups (Table 8). Principal coordinate analysis (PCoA) (Figure 1b) showed that the AMPs group was more concentrated in the plot, and the microbial composition of the samples was the closest. Regarding intergroup differences, the AMPs and TA groups were clustered in the graph, indicating a higher similarity in microbial composition. The unweighted pair group method with arithmetic means (UPGMA) result agreed with the PCoA results shown in Figure 1c.

At the phylum level, the relative abundance of Firmicutes was the highest in the fecal microbial community, followed by Bacteroidetes and Spirochaetes (Figure 2a). Compared to that in the ZnO group, the relative abundance of Spirochaetes in the fecal microorganisms of the TA Group piglets was higher, and the quantity of Cyanobacteria in the fecal microorganisms of the AMPs group piglets was higher (*p* < 0.05) (Table 9). At the family level, the average abundance of piglet fecal microbial community was more than 1%, including the *Ruminococcaceae*, *Prevotellaceae*, *Muribaceae*, *Lactobacilliaceae*, *Lachnospiriceae*, and *Streptococcaceae*, etc (Figure 2b). The relative abundance of the *Rikenellaceae* in the feces of piglets in the ZnO and TA groups was higher than that of piglets in the AMPs (*p* < 0.05) (Table 10). The top-10 abundance taxa at the genus level are shown in Figure 2c. *Uncultured_bacterium f Muribaculaceae* showed the highest relative abundance in the fecal samples (Table 11). The relative abundance of the genus *Treponema_2* of weaned piglets in the TA group was higher than that in the ZnO group (*p* < 0.05) and not significantly different from that of the AMPs group (*p* > 0.05). The abundance of the *Prevotellaceae NK3B31* group decreased in the feces of piglets in the AMPs and TA group compared with that in the ZnO group (*p* < 0.05), while the abundance of *Lachnospiraceae* increased (*p* < 0.05). The relative abundance of *Treponema_2* in the feces of piglets in the ZnO, AMPs, and TA groups increased successively (*p* < 0.05). There were no significant (*p* > 0.05) changes in the abundance of other genera of fecal microorganisms among the three groups.

## 4. Discussion

Lehrer et al. [29] and Viljanen et al. [30] found that human defensins enhanced the outer membrane permeability of *E. coli* ML-35 and *Salmonella typhimurium*, thereby affecting phage membrane function and achieving bacteriostatic effects, respectively. Some AMPs can inhibit the growth of *S. aureus* by inhibiting adhesion and biofilm formation [31]. Costabile et al. [32] indicated that 1‒6 mg/mL TA strongly inhibits *Salmonella typhimurium* and is a promising livestock feed additive. However, Parys et al. [33] injected piglets with 107 units of *Salmonella*, fed them 3 mg/kg TA per day, and found that TA had no significant effect on *Salmonella* in the feces and intestines of piglets. These inconsistent results may be attributed to species differences and the amount of TA used. In our study, the antibacterial test results showed that the MIC of AMPs against *E. coli* and *Salmonella* were 4 and 2 mg/mL, respectively, which were larger than those of ZnO and TA. The MIC of TA against *Salmonella* was 0.5 mg/mL, smaller than that of ZnO and AMPs. The results of the MIC suggested that the bacteriostatic effects of TA, ZnO, and AMPs decreased successively. However, the antibacterial effects of TA, ZnO, and AMPs are related to their sources and doses, and it is impossible to determine what works best and is more rigorous.

Previous studies have shown that ZnO, AMPs, and TA can improve the growth performance of organisms [5,34,35]. Jensen-Waern et al. [36] reported that adding 2500 mg/kg of ZnO to feed increased the average daily weight gain of weaned piglets by 10‒16% and partially increased feed conversion. Xiong et al. [37] found that adding AMPs to feed significantly increased ADG and ADFI in weaned piglets. TA has a strong affinity for certain nutrients and is generally considered to negatively affect monogastric animals’ growth performance and nutrient digestibility [38]. However, 0–1% TA supplementation did not adversely affect growth performance and nutrient digestibility in monogastric animals has been reported [39]. This study showed TA and AMPs had the same effect as ZnO on the growth performance of the weaned piglets. The possible reason was that AMPs and TA improve the growth performance through decreased diarrhea and increased nutrient digestibility. The occurrence of diarrhea can elevate piglet mortality rates and impede their growth performance [40]. Previous studies have shown that pharmacological doses of ZnO can effectively reduce diarrhea caused by weaning stress in piglets [41,42]. Some AMPs inhibit the proliferation of harmful bacteria through electrostatic interactions with lipopolysaccharides, maintain intestinal microecology, and reduce the incidence of diarrhea [43]. In addition, TA forms a membrane in the intestinal wall and reduces intestinal irritation acting as an anti-diarrheal agent [44]. The polyphenol hydroxyl structure gives TA good physiological functions such as convergence, antidiarrhea, antibacterial, antioxidant, and antiviral activities [45]. Our study found that there was no significant difference in the diarrhea rate of weaned piglets among the three groups. In other words, ZnO, AMPs, and TA have similar anti-diarrheal effects.

Previous studies have shown that ZnO, AMPS, and TA have beneficial effects on the nutrient digestibility of animals. The digestibility of dry matter and crude protein in broiler chickens increases linearly with increasing zinc concentration in the daily diet [46]. This may be due to Zn being a cofactor for certain digestive enzymes. It appears that dietary AMP supplements increased the nutrient digestibility of piglets mainly through their beneficial effects on gut microecology [47]. The digestibility of nutrients in weaned piglets fed a diet containing potato AMPs was the same as that of piglets fed on an antibiotic diet, and both were higher than that in the control group [48]. This indicated that the ability of potato proteins to improve the digestibility of nutrients was at the same level as antibiotics. In addition, Schiavone et al. [35] showed that adding TA to the diet reduced intestinal peristalsis and slowed the passage of chyme through the small intestine, thereby increasing broiler feed digestibility and significantly increasing ADG and ADFI in broiler chickens. In this study, the digestibility of dry and organic matter and crude fat of piglets in the AMPs and TA groups was higher than that of piglets in the ZnO group. The apparent tract digestibility of nutrients in piglets fed diets supplemented with the AMPs and TA better than ZnO might be due to the modulation of gut environment and improvement of intestine microbial balance [49,50]. On the other hand, the improvement in nutrient digestibility by TA has also been shown to be associated with the stimulation of saliva and bile secretion, as well as the induced enhancement of enzyme activity [51]. The increase in plasma IgA, IgM, and IgG partly reflects the enhanced humoral immune function [52]. Chang et al. [53] reported that ZnO was beneficial in improving calves’ serum IgG content and immune function. However, there is also evidence that high zinc levels inhibit immune function in the body [54]. Therefore, whether proper zinc supplementation can have a positive impact on immunity is still uncertain. Wu et al. [14] found that adding AMPs to the diet increased piglet serum immunoglobulin and cytokine levels. In addition, Ramah et al. [55] reported that high doses of 30 g/kg TA inhibited spleen humoral immunity and cytokine mRNA expression, weakening the immune response, while a low dose of 0.5 g/kg TA enhanced the humoral immune response and improved health. In this study, the plasma IgM and IL-6 contents of piglets in the ZnO and AMPs groups on day 28 were higher than that of the piglets in the TA group. IgM is the first antibody produced in an antigen-stimulated fluid immune response and plays an important role in the early defense of the body [56]. Therefore, results suggested that ZnO and AMPs have a stronger regulatory effect on the immune function of weaned piglets than TA. The reason why TA is less effective in immunization than ZnO and AMPs may be related to the dose and type of addition. Excessive free radicals produced by weaning stress can damage biomolecules and cause piglet tissue and cell damage [57]. The antioxidant defense system of piglets eliminates free radicals derived from oxidative metabolism. T-AOC, T-SOD, GSH-PX, and CAT are important enzymes of the antioxidant system, and their activities are directly proportional to the body’s ability to resist free radicals [58]. MDA is a metabolite of lipid peroxidation in the cell membrane, and its concentration reflects the degree to which the cell membrane is oxidized by oxidative tissue damage [59]. Previous studies have shown that ZnO, AMPs, and TA play important regulatory roles in the antioxidant defense system [60,61,62]. In this study, no significant differences were observed in the activities of antioxidant enzymes (T-AOC, SOD, GSH-PX, and CAT) or MDA concentrations among the three groups. These results indicate that AMPs and TA have similar regulatory functions in the antioxidant system of weaned piglets as ZnO.

The hindgut of pigs contains dense and metabolically active microbiota, consisting mainly of bacteria, which affects host nutrition, immunity, and physiological processes [63]. Studies have shown that the higher the diversity of gut flora in animals, the more balanced the gut intestinal microecology [64,65]. In this study, the alpha diversity results showed that the Chao1 and ACE indices of fecal microorganisms of piglets in the AMPS group were significantly higher than those of the ZnO and TA groups. This suggests that microbiome abundance in the AMPs group was higher than that in the ZnO and TA groups. Pharmacological doses of ZnO have been shown to reduce microbial abundance in the ileal chyme [66]. AMPs have also been reported to reduce the number of harmful bacteria in intestinal digests and increase the number of beneficial bacteria [67]. The natural polyphenolic structure of TA enables it to selectively inhibit the growth of harmful bacteria, indirectly promote the proliferation of beneficial flora, and regulate colony balance [68]. *Treponema_2* is one of the cellulolytic bacteria [69]. Wenner et al. [70] have found that supplementation of Zn decreased the relative abundance of *Treponema_2*. This may be the reason why the relative abundance of *Treponema_2* was lower in the ZnO group than in the TA groups. This may also be one of the reasons why the nutrient digestibility of piglets in the TA groups was higher than that of the ZnO group. In addition, at the genus level, the relative abundance of *Lachnospiraceae* in the feces of the AMPs and TA groups was higher than that of piglets in the ZnO group. *Lachnospiraceae* hydrolyze starch and other sugars to produce butyrate and other short-chain fatty acids in the porcine intestinal tract to provide energy [71]. These results indicate that the supplementation of AMPs and TA in weaned piglet diets is more beneficial in producing short-chain fatty acids than ZnO. The above results indicate that adding TA and AMPs to the diet can effectively increase the number of beneficial bacteria and improve the intestinal microbial homeostasis.

## 5. Conclusions

The MIC of TA in vitro was lower than those of ZnO and AMPs. Compared to ZnO, AMPs increased the digestibility of crude fat, and AMPs and TA increased the digestibility of dry and organic matter. There were no differences in the diarrhea rate and antioxidant ability of piglets in the ZnO, AMPs, and TA groups. The plasma IgM and IL-6 contents of piglets in the TA group were lower than that of piglets in the ZnO and AMPs groups. The addition of AMPs and TA to the diet also increased the number of beneficial bacteria in the intestine compared to the addition of ZnO. AMPs and TA can be added to the diets of weaned piglets as substitutes for ZnO to reduce diarrhea, improve growth, and promote the intestinal health of weaned piglets.

## Figures and Tables

**Figure 1 animals-13-01797-f001:**
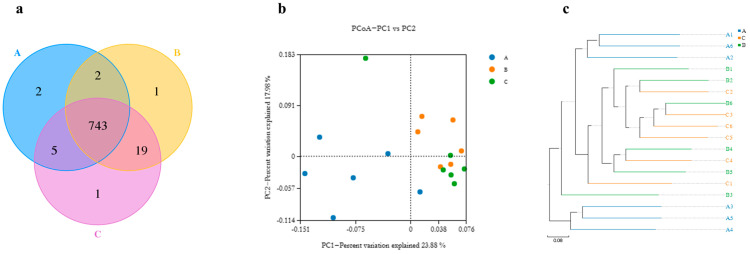
Veen diagram of fecal microorganisms in piglets (**a**), PCoA analysis of piglet fecal microbes (**b**), and UPGMA analysis of piglet fecal microbes (**c**). A, zinc oxide; B, antimicrobial peptides; C, tannic acid.

**Figure 2 animals-13-01797-f002:**
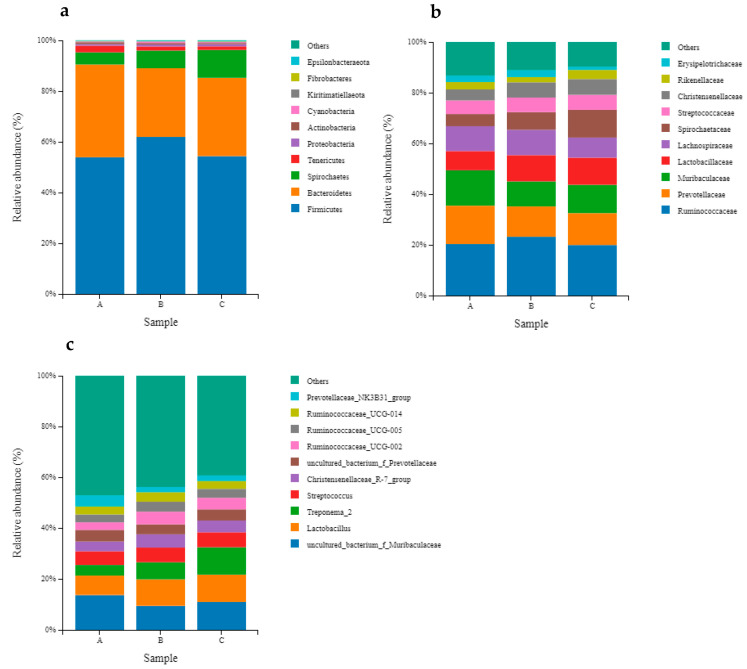
The relative abundance of fecal microbiota of piglets at phylum (**a**), family (**b**) and genus (**c**) levels. A, Zinc Oxide; B antimicrobial peptides; C, tannic acid.

**Table 1 animals-13-01797-t001:** The ingredients and nutritional composition of diets (DM, %).

Ingredients	Content	Nutritional Composition	Content
Corn	66.34	DE ^2^ (MJ/kg)	14.51
Soybean meal	9.00	CP	18.73
Fish meal	5.00	Ca	0.83
Whey	4.60	Total P	0.69
Soybean oil	3.00	Crude fat	3.43
Soy protein powder	9.00		
CaHPO_4_	0.40		
CaCO_3_	0.70		
NaCl	0.23		
L-Lysine HCl	0.36		
Met	0.20		
Thr	0.15		
Trp	0.02		
Trace mineral premix ^1^	1.00		
Total	100.00		

^1^ The premix provided the following per kg of diets: V_A_ 10,500 IU, V_D3_ 3000 IU, V_E_ 22.5 IU, V_K3_ 3.0, mg, pantothenic 15 mg, riboflavin 7.5 mg, folic acid 1.5 mg, niacinamide 30.0 mg, thiamine 3.0 mg, V_B6_ 4.5 mg, biotin 0.12 mg, V_B12_ 0.03 mg, Zn 100 mg, Fe 100 mg, Mn 4.0 mg, Cu 6.0 mg, I 0.3 mg, Se 0.3 mg. ^2^ Digestive energy is calculated value. Abbreviations: CaHPO_4_, Calcium hydrogen phosphate; CaCO_3_, Calcium carbonate.

**Table 2 animals-13-01797-t002:** Minimum inhibitory concentration of zinc oxide, antibacterial peptide, and tannic acid on indicator bacteria.

Items		Treatments (mg/mL)					MIC (mg/mL)
	0	0.5	1	2	4	8	
**ZnO**							
*Escherichia coli*	—		10.5 ± 0.38	11.3 ± 0.25	12.4 ± 0.42	13.0 ± 0.20	1
*Staphylococcus aureus*	—		10.1 ± 0.51	10.3 ± 0.34	11.0 ± 0.66	12.0 ± 0.72	1
*Salmonella*	—		10.7 ± 0.30	11.1 ± 0.57	11.3 ± 0.40	12.3 ± 0.80	1
**AMPs**							
*Escherichia coli*	—		—	—	10.3 ± 0.19	12.2 ± 0.87	4
*Staphylococcus aureus*	—		10.3 ± 0.49	11.6 ± 0.27	12.8 ± 0.45	13.9 ± 2.91	1
*Salmonella*	—		—	11.5 ± 0.24	12.2 ± 0.10	12.4 ± 1.43	2
**TA**							
*Escherichia coli*	—		10.3 ± 0.76	11.2 ± 0.08	12.4 ± 0.49	13.5 ± 0.41	1
*Staphylococcus aureus*	—		10.1 ± 0.59	10.4 ± 0.28	11.8 ± 0.33	12.5 ± 0.77	1
*Salmonella*		10.6 ± 0.51	11.1 ± 1.36	10.9 ± 0.57	11.5 ± 0.23	12.3 ± 0.94	0.5

Abbreviations: AMPs, antimicrobial peptides; TA, tannic acid; ZnO, zinc oxide; MIC, minimum inhibitory concentration. —, No antibacterial effect.

**Table 3 animals-13-01797-t003:** Effects of antibacterial peptides and tannic acid as the substitutes for zinc oxide on growth performance, apparent nutrient digestibility, and diarrhea of weaned piglets.

**Items**	**Treatments**			**SEM**	***p*-Value**
	**ZnO**	**AMPs**	**TA**		
Initial weight (kg)	7.42	7.42	7.41	0.16	0.999
Final weight (kg)	15.7	16.2	16.0	0.23	0.717
Average daily feed intake (g)	296	314	307	5.29	0.419
Average daily body gain (g)	530	534	531	1.18	0.484
Feed intake/body weight gain	1.80	1.72	1.73	0.03	0.823
Diarrhea rate (%)	5.57	11.1	5.00	2.15	0.447
**Apparent nutrient digestibility (%)**					
DM	86.0 ^b^	88.0 ^a^	88.2 ^a^	0.33	0.006
CP	79.4	81.4	80.6	0.62	0.251
Organic matter	88.9 ^b^	90.4 ^a^	90.7 ^a^	0.26	0.004
Ca	68.1	71.7	70.5	0.75	0.074
P	67.8	68.9	70.6	0.78	0.374
Crude fat	70.0 ^b^	78.3 ^a^	75.8 ^ab^	1.50	0.034

^a,b^ Values within a row with different superscripts differ significantly (*p* < 0.05). Abbreviations: ZnO, zinc oxide; AMPs, antimicrobial peptides; TA, tannic acid.

**Table 4 animals-13-01797-t004:** Effects of antibacterial peptides and tannic acid as the substitutes for zinc oxide on plasma biochemical indicators of weaned piglets.

Items	Treatments			SEM	*p*-Value
	ZnO	AMPs	TA		
**Day 14**					
ALP (U/L)	1.88	2.32	2.74	0.53	0.822
AST (U/L)	4.90	5.23	5.41	0.81	0.969
ALT (U/L)	17.1	16.2	14.1	1.77	0.796
GH (pg/mL)	2.89	2.90	3.05	0.15	0.891
**Day 28**					
ALP (U/L)	0.95	2.09	1.62	0.34	0.407
AST (U/L)	8.72	10.3	6.62	1.01	0.408
ALT (U/L)	17.2 ^a^	6.5 ^b^	16.8 ^a^	1.66	0.004
GH (pg/mL)	3.18 ^a^	2.72 ^ab^	2.62 ^b^	0.10	0.027

^a,b^ Values within a row with different superscripts differ significantly (*p* < 0.05). Abbreviations: ALP, alkaline phosphatase; AST, aspartate aminotransferase; ALT, glutamic pyruvic transaminase; GH, growth hormone; ZnO, zinc oxide; AMPs, antimicrobial peptides; TA, tannic acid.

**Table 5 animals-13-01797-t005:** Effects of antibacterial peptides and tannic acid as the substitutes for zinc oxide on plasma immunoglobulin of weaned piglets.

Items	Treatments			SEM	*p*-Value
	ZnO	AMPs	TA		
**Day 14**					
IgA (μg/mL)	6.02	5.87	5.41	0.20	0.457
IgG (μg/mL)	61.9	67.1	71.3	2.52	0.334
IgM (μg/mL)	6.85	6.44	5.92	0.27	0.390
**Day 28**					
IgA (μg/mL)	6.08	5.90	6.99	0.36	0.434
IgG (μg/mL)	72.9	63.2	72.8	2.21	0.119
IgM (μg/mL)	6.50 ^a^	6.32 ^a^	5.13 ^b^	0.23	0.018

^a,b^ Values within a row with different superscripts differ significantly (*p* < 0.05). Abbreviations: Ig, immunoglobulins; ZnO, zinc oxide; AMPs, antimicrobial peptides; TA, tannic acid.

**Table 6 animals-13-01797-t006:** Effects of antibacterial peptides and tannic acid as the substitutes for zinc oxide on plasma cytokines of weaned piglets.

Items	Treatments			SEM	*p*-Value
	ZnO	AMPs	TA		
**Day 14**					
IL-2 (pg/mL)	139	134	154	5.58	0.325
IL-6 (pg/mL)	34.0 ^b^	42.6 ^ab^	47.6 ^a^	2.26	0.033
IL-8 (pg/mL)	83.1	77.7	86.1	2.07	0.262
**Day 28**					
IL-2 (pg/mL)	150	140	143	5.81	0.758
IL-6 (pg/mL)	27.0 ^a^	28.7 ^a^	22.1 ^b^	1.19	0.030
IL-8 (pg/mL)	86.0	85.7	93.9	2.04	0.209

^a,b^ Values within a row with different superscripts differ significantly (*p* < 0.05). Abbreviations: IL, interleukin; ZnO, zinc oxide; AMPs, antimicrobial peptides; TA, tannic acid.

**Table 7 animals-13-01797-t007:** Effects of antibacterial peptides and tannic acid as the substitutes for zinc oxide on plasma antioxidant capacity of weaned piglets.

Items	Treatments			SEM	*p*-Value
	ZnO	AMPs	TA		
**Day 14**					
T-AOC (mM)	0.39	0.34	0.35	0.01	0.208
CAT (U/mL)	8.17	6.76	9.37	0.73	0.362
T-SOD (U/mL)	35.0	41.5	35.0	2.38	0.477
GSH-Px (U/mL)	34.2	34.0	40.8	1.98	0.296
MDA (nmol/L)	5.35	4.98	5.00	0.07	0.052
**Day 28**					
T-AOC (mM)	0.44	0.39	0.35	0.02	0.065
CAT (U/mL)	6.08	5.96	3.65	0.58	0.159
T-SOD (U/mL)	41.5	42.5	50.2	2.61	0.366
GSH-Px (U/mL)	66.6	67.1	67.6	2.09	0.983
MDA (nmol/L)	5.21	5.04	4.88	0.08	0.227

Abbreviations: T-AOC = total antioxidant capacity; CAT = catalase; T-SOD = total superoxide dismutase; GSH-Px = glutathione peroxidase; MDA = malondialdehyde; ZnO, zinc oxide; AMPs, antimicrobial peptides; TA, tannic acid.

**Table 8 animals-13-01797-t008:** Effects of antibacterial peptides and tannic acid as the substitutes for zinc oxide on alpha diversity of fecal microorganisms of weaned piglet.

Items	Treatments			SEM	*p*-Value
	ZnO	AMPs	TA		
Chao1	648 ^b^	702 ^a^	680 ^ab^	8.49	0.021
ACE	657 ^b^	716 ^a^	689 ^ab^	8.99	0.017
Shannon	6.79	6.88	6.75	0.06	0.693
Simpson	0.97	0.98	0.97	0.002	0.701

^a,b^ Values within a row with different superscripts differ significantly (*p* < 0.05). Abbreviations: ZnO, zinc oxide; AMPs, antimicrobial peptides; TA, tannic acid.

**Table 9 animals-13-01797-t009:** Effects of antibacterial peptides and tannic acid as the substitutes for zinc oxide on the fecal microbial abundance of weaned piglet at the phylum level (%).

Items	Treatments			SEM	*p*-Value
	ZnO	AMPs	TA		
Firmicutes	54.0	61.8	54.3	1.96	0.182
Bacteroidetes	36.6	27.2	30.9	1.71	0.067
Spirochaetes	4.73 ^b^	6.96 ^ab^	11.1 ^a^	1.03	0.027
Tenericutes	2.62	1.62	1.42	0.24	0.074
Proteobacteria	0.59	0.75	0.84	0.09	0.511
Actinobacteria	0.77	0.61	0.49	0.06	0.173
Cyanobacteria	0.24 ^b^	0.57 ^a^	0.35 ^ab^	0.06	0.037

^a,b^ Values within a row with different superscripts differ significantly (*p* < 0.05). Abbreviations: ZnO, zinc oxide; AMPs, antimicrobial peptides; TA, tannic acid.

**Table 10 animals-13-01797-t010:** Effects of antibacterial peptides and tannic acid as the substitutes for zinc oxide on the fecal microbial abundance of weaned piglet at the family level (%).

Items	Treatments			SEM	*p*-Value
	ZnO	AMPs	TA		
*Ruminococcaceae*	20.3	23.1	19.9	0.90	0.291
*Prevotellaceae*	15.3	12.0	12.6	1.12	0.466
*Muribaculaceae*	13.9	9.90	11.2	0.83	0.137
*Lactobacillaceae*	7.60	10.4	10.6	0.85	0.273
*Lachnospiraceae*	9.79	9.98	7.92	0.79	0.530
*Streptococcaceae*	5.37	5.72	5.88	0.64	0.952
*Christensenellaceae*	4.32	5.99	6.20	0.60	0.392
*Rikenellaceae*	2.94 ^ab^	2.12 ^b^	3.64 ^a^	0.25	0.029
*Uncultured_bacterium-o-Mollicutes RF39*	2.62	1.61	1.41	0.24	0.072
*Erysipelotrichaceae*	2.47	2.65	1.22	0.12	0.092
*Tannerellaceae*	1.40	1.04	0.88	0.16	0.171
*Acidaminococcaceae*	1.53 ^a^	1.22 ^ab^	0.49 ^b^	0.23	0.017
*VeillonellaceaeFamily_XIII*	0.85	1.21	0.36	0.09	0.333

^a,b^ Values within a row with different superscripts differ significantly (*p* < 0.05). Abbreviations: ZnO, zinc oxide; AMPs, antimicrobial peptides; TA, tannic acid.

**Table 11 animals-13-01797-t011:** Effects of antibacterial peptides and tannic acid as the substitutes for zinc oxide on the microbial abundance of weaned piglet at the genus level (%).

Items	Treatments			SEM	*p*-Value
	ZnO	AMPs	TA		
*Uncultured_bacterium f Muribaculaceae*	13.6	9.47	10.9	0.47	0.100
*Lactobacillus*	7.55	10.4	10.6	0.21	0.273
*Treponema_2*	4.10 ^b^	6.73 ^ab^	10.8 ^a^	0.44	0.021
*Streptococcus*	5.37	5.72	5.88	0.26	0.952
*Uncultured bacterium f Prevotellaceae*	4.66	3.94	4.42	0.22	0.734
*Christensenellaceae R-7 group*	3.78	5.19	4.61	0.24	0.535
*Ruminococcaceae_UCG-002*	2.93	4.97	4.50	0.25	0.179
*Ruminococcaceae_UCG-005*	3.07	3.92	3.51	0.27	0.276
*Prevotellaceae NK3B31 group*	4.37 ^a^	2.06 ^b^	2.13 ^b^	0.30	0.038
*Lachnospiraceae*	0.63 ^b^	3.79 ^a^	3.51 ^a^	0.26	0.009
*Rikenellaceae RC9 gut group*	2.62	1.85	3.05	0.24	0.092
*uncultured_bacterium_f Lachnospiraceae[Eubacterium]*	2.97	2.64	2.24	0.20	0.529
*Coprostanoligenes group*	2.46	2.06	1.55	0.10	0.144

^a,b^ Values within a row with different superscripts differ significantly (*p* < 0.05). Abbreviations: ZnO, zinc oxide; AMPs, antimicrobial peptides; TA, tannic acid.

## Data Availability

The rest of the raw data supporting the conclusions of this article will be made available by the authors without any undue reservation.

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
