# Peer review of "Supplementation with Antimicrobial Peptides or a Tannic Acid Can Effectively Replace the Pharmacological Effects of Zinc Oxide in the Early Stages of Weaning Piglets"

_animals, 2023, doi:10.3390/ani13111797_

Round 1

Reviewer 1 Report

Isolated insect bioactive compounds and their potential use in animal diets and medicine

Dear Authors,

The manuscript is interesting and describes  solutions important from practical and environmental point of view. The most important is that the tannic acid and antimicrobial peptides gives promising results. In the experiment didn’t observed significant difference in performance of piglets and plasma biochemical indicators, except level of the growth hormone. But its level was enough to obtain the same growth rate.

Manuscript is very well prepared and there are not many elements to correct. Below I added some suggestions helpful during this process (there was a small problem during review because in pdf file I didn’t obtain added lines, but changes and suggestions are mainly related with tables and references) :

Line 120

Table 1

Description in case of the digestible energy should be in the same line: DE2 (MJ/kg).

Line 177

Table 2

Maybe in case of AMPs and TA is better to use bold, because every treatment will be more visible or it could be also separated a horizontal line, and centred.

Line 190

Table 3 from the end of manuscript must be paste in place of table 4 in subsection 3.2 Growth performance/ Performance, and table 4 must be paste in subsection 3.3 Blood Parameters about line 203

In table 4:  ALT (U/L) in case of AMPs, shouldn’t be 16.5 instead of 6.5? (because between the other indicators there is no such big differences).

Line 207, 216, 222

Respectively Table 5, 6, 7

The same like in line 177

Line 263

Figure 2: labels with percentages (or information about values in chart legend) could gives more precise information about microbial content in faeces in comparison between each treatment.

Line 266

Table 9

In case of SEM for Cyanobacteria it is possible to change numbers of decimals for three (p = 0,000 or p = 0,001, it depends from 5th decimal number)

Line 270

Table 10

Family XIII, SEM can be changed for 0,001

Line 400

In manuscript is ‘…promotes Lactobacillus proliferation…’, perhaps there should be Lactobacillus sp.

± line 433

References:

·       35 (Van Parys et al. 2010) – Journal abbreviation needed

·       37 (Bechinger et al. 2017) – two dots in Journal title needed and spaces between volume and page numbers

·       40 (Stukelj et al. 2010) – the same like in 37 and italic in case of Journal title

·       42-46 – italic (Journal title)

·       76 - two dots in Journal title needed

Table 3 must be transferred to line 190

Reviewer 2 Report

This paper present results from an experiment in which thirty piglets were randomly allotted to one of three dietary treatments, ZnO (2000 mg/kg ZnO), AMPs (700 mg/kg AMPs), and TA (1000 mg/kg TA diet), to investigated whether AMPs and TA could effectively replace the effects of pharmacological doses of ZnO on weaned piglets in the same growth environment. The authors compared growth performance, gut microbes, blood biochemistry, and other indicators of weaning piglets.

The experimental design and the methods applied are clearly described. However, the manuscript often takes the form of a Review and not of a Research paper. From my point of view, the data obtained from this experimentation should be commented in more depth, trying to give a better explanation of the possible differences with the data present in the literature.

in page 6, line 4 of "3.2.Growth performance" paragraph, replace P < 0.05 with P > 0.05

As underlined by my comments, it is clear that I consider the paper consistent with the journal but that it should be reviewed by the authors, commenting and justifying the results obtained in a more detailed way, not making a simple comparison with the results obtained from other scientific works.

As for English, I haven't noticed any particular grammatical errors. The conclusions seem adequate.

From my point of view, after the suggested insights, the work could be considered.

Reviewer 3 Report

The authors present a well written manuscript describing dietary replacements for Zinc Oxide (ZnO) in piglet feeds. The underlying research question is whether ZnO can be replaced with Tannic acids (TA) or antimicrobrial peptides (AMPs) in weaning piglet diets.  Due to the global interest in finding alternatives to ZnO, this research is of particular interest. The authors conclude that TA and AMPs can increase digestibility of certain nutrients compared to ZnO, that there was no difference in diarrhea rate of piglets and that TA and AMPs promote the proliferation of beneficial micro-organisms in the gut. 

I believe the manuscript would be improved following revisions and therefore recommend the manuscript for publication after revision. 

Abstract comments: 

1. I suggest the title is slightly revised to "Supplementation with antimicrobrial peptides or tannic acids can effectively replace the pharmacological effects of zinc oxide in the early stages of weaning piglets"

2. I believe there is a typo "The crude fat and TA significantly (P < 0.05) increased the digestibility of dry and organic matter". 

3. I think that replacements is a stronger word than substitutes  "added to feed as substitutes for ZnO". Since ZnO is being phased out of animal feeds, this emphasises the movement away from reliance on ZnO. 

Intro comments: 

1. I believe there is a typo, should this be mg/kg of feed?  "Zinc oxide (ZnO) given at a pharmacologic dose of 1500‒2500 mg/kg of Zn has antibacterial and growth-promoting properties that increases the body's antioxidant capacity and reduces the occurrence of withdrawal stress."

2. Do the authors mean anti-nutritional factors? "Some amounts of TA are considered antinutrient substances because they precipitate proteins, inhibit digestive enzymes, and affect the utilization of vitamins and minerals [18]."

Materials and methods comments: 

1. AMPs are already defined, no need to use the full word "Antimicrobial peptides (AMPs) and TA suspensions were prepared using the same procedure."

2. In section 2.3, it would be good if the authors could elaborate on the source of TA and the source and type of AMP. 

3. I suggest rephrasing "including an adaptation period of 1‒7 days." as it is slightly confusing. Perhaps swapping of 1-7 days for, "for the first 7 days of the experiment". 

4. I suggest rephrasing "The piglets were weighed before feeding on days 1 and 29 (at the end of the experiment)." Perhaps, "The piglets were weighed at the start (day 1) and end (day 29) of the experiment. 

5. It seems that the pigs were housed individually and feed offered and refused was recorded daily. Did the authors consider analysing the observed daily feed intakes? 

Results comments: 

1. The highlighted area can be deleted "The MIC of TA against E. coli and S. aureus was 1 mg/ml, and the MIC against E. coli, S. aureus, and Salmonella were 1, 1, and 0.5 mg/ml, respectively."

2. Table 3 was placed at the back of the manuscript. Could the authors check the statistics for diarrhea rate? It seems surprising that there is no significant difference given the SEM and mean values. 

3. Table 3 states that n=6, but in the MM there are 10 pigs per treatment. Can the authors provide an explanation for this? 

4. A general comment across all tables, please take care and check formatting for boldface (e.g. D14 and D28) and border lines. 

5. Figure 2 legend, suggest changing to "The relative abundance of faecal microbiota of piglets at phylum (a), family (b) and genus (c) levels. A, Zinc Oxide; B antimicrobrial peptides; C, tannic acid.

6. Tables 9-11 appear to present the same information as Figure 2, is it necessary to present both the tables and the figure? 

Discussion comments:

1. Could the authors elaborate on the results from reference 40 "However, the effects of a 0‒1% TA supplementation on growth performance and nutrient digestibility in monogastric animals have been reported [40]."

2. Very minor suggestion but keep the order of ZnO, AMPs and TA the same in "This study showed no significant difference in the growth performance of piglets fed diets containing ZnO, AMPs, or TA [6]; ZnO, TA, and AMPs had the same effect on the growth performance of the weaned piglets."

3. Suggest that the authors elaborate on "The digestibility of nutrients in weaned piglets fed a diet containing potato protein was the same as that of piglets fed on an antibiotic diet [44].". At the moment it is not clear the relevance of the sentence. 

4. Suggest changing "speculated" for "results suggested" or something similar in "Therefore, it is speculated that ZnO and AMPs have a stronger regulatory effect on the immune function of weaned piglets than TA."

5. Typo "Some mineral elements have also been reported to reduce the number of harmful bacteria in intestinal digests and increase the number of beneficial bacteria [60]."
